# Clinical Outcome of Cefiderocol for Infections with Carbapenem-Resistant Organisms

**DOI:** 10.3390/antibiotics12050936

**Published:** 2023-05-21

**Authors:** Monirul I. Sajib, Melinda Monteforte, Roderick Go

**Affiliations:** Division of Infectious Disease, Department of Medicine, Stony Brook University, Stony Brook, NY 11794, USA

**Keywords:** cefiderocol, gram-negative infection, multidrug-resistant infection

## Abstract

Cefiderocol is a novel cephalosporin recently approved by the FDA to aid clinicians in the fight against multidrug-resistant (including carbapenem-resistant) gram-negative organisms. The primary objective of this study is to evaluate the 14- and 28-day mortality associated with cefiderocol. We performed a retrospective chart review of all adult patients admitted at Stony Brook University Hospital between October 2020 and December 2021 and received cefiderocol for at least 3 days. Patients were excluded if they received more than one course of cefiderocol therapy or remained hospitalized at the time of this study. A total of 22 patients met the inclusion criteria. The all-cause mortality on day 28 for all patients was 13.6%, whereas this rate for patients with BSI was 0%, with cUTI was 0% and with LRTI was 16.7%. The all-cause mortality on day 28 for patients who received the dual antibiotics (in conjunction with cefiderocol) was 0%, compared to 25% for patients who only received cefiderocol (*p* = 0.25). We noted treatment failure in two patients (9.1%). Our findings suggest that cefiderocol could possibly be associated with lower all-cause mortality than previously thought. In our study, we did not find any significant difference between cefiderocol’s use in combination with another antibacterial agent and its use as a monotherapy.

## 1. Introduction

Infections due to carbapenem-resistant Enterobacterales (CRE), carbapenem-resistant *Acinetobacter baumannii* (CRAB), difficult-to-treat resistant *Pseudomonas aeruginosa* (DTR-P) and *Stenotrophomonas maltophilia* are of great concern due to the limited effective antimicrobial options. There are only a few treatments options currently available for treatment of infections with such organisms, i.e., colistin, ceftazidime-avibactam and ceftolozane-tazobactam. Drug resistance was previously been reported with these agents, which raises the need for studying newer antibiotic options [1,2,3,4,5,6].

Cefiderocol is a novel cephalosporin that is currently FDA-approved for the treatment of patients with serious gram-negative bacterial infections [7,8]. Cefiderocol has potent in vitro efficacy against CRE, CRAB, DTR-P and *S. maltophilia* [3,6,9,10]. However, the current available data on clinical outcome and efficacy of cefiderocol are limited. There are concerns about the utility of cefiderocol in patients with bloodstream infections (BSI), ventilator-associated bacterial pneumonia (VABP) and hospital-acquired bacterial pneumonia (HABP), as an increase in all-cause mortality was reported with these types of infections [11,12,13]. Given the limited treatment options available for drug-resistant gram-negative bacterial infections, we performed this retrospective cohort study to assess the outcomes and mortality in patients treated with cefiderocol.

## 2. Results

A total of 22 patients met the inclusion criteria. Demographic data is listed in Table 1. The average age of the patients was 63.6 years. In total, 15 patients (68.2%) were male, and 7 (31.8%) patients were female. Their average Charlson comorbidity index (CCI) was 5.4, which corresponds to an estimated 10-year survival between 2% and 21%. The range of length of stay was between 11 and 220 days, with an average stay of 53 days. A total of 6 patients had lower respiratory tract infections (LRTI), 3 patients had bloodstream infections, 2 patients had complicated urinary tract infections (cUTI) and 11 patients had infections at other sites (Table 2). Infections at other sites included intrabdominal, orbital and buccal abscess, sacral, lower extremity and facial wounds. The types of gram-negative bacterial isolates identified are listed in Table 2. These results include 14 isolates of *P. aeruginosa*, 8 isolates of *A. baumannii* complex and 1 isolate of *S. maltophilia*.

The all-cause mortality on day 14 and day 28 for all patients were found to be 4.5% and 13.6%, respectively. When sub-analysis of the all-cause mortality was performed based on the types of infections, it was found to be 0% on both day 14 and day 28 for patients with cUTI. In patients with bloodstream infections, the all-cause mortality was also found to be 0% on both day 14 and day 28. However, the all-cause mortality on day 14 and day 28 for patients with lower respiratory tract infections was found to be higher, being 16.7% (1 of 6 patients died) at both points. Notably, CRAB was isolated from the sputum culture of patients’ who died from lower respiratory tract infection (VABP). Furthermore, none of the patients in this cohort were infected with SARS-CoV-2 during the hospitalization when cefiderocol was prescribed. Finally, the all-cause mortality on day 14 and day 28 for patients with infections at other sites were 0% and 18.2% (2 of 11 patients died), respectively. Of these two patients, one had a sacral wound with DTR-P, for which debridement was performed 4 days prior to cefiderocol therapy, and the other had an intra-abdominal abscess with DTR-P, for which drainage was performed six days prior to cefiderocol therapy. These data are summarized in Table 3.

In addition, we analyzed the all-cause mortality for patients who received combination therapy with an additional gram-negative antibiotic along with cefiderocol, and compared these data with those of patients who only received cefiderocol (Table 4). The antibiotics used in combination with cefiderocol were intravenous aminoglycosides (gentamicin, tobramycin), tetracyclines (tigecycline, eravacycline, omadacycline) and fluroquinolones (ciprofloxacin). We found that the all-cause mortality on day 14 for patients who received combination therapy was 0% versus 8.3% for those who only received cefiderocol (*p* = 1.0). Similarly, the all-cause mortality on day 28 for patients who received combination therapy was 0%, compared to 25% for patients who only received cefiderocol (*p* = 0.25).

We also calculated the all-cause mortality based on the types of gram-negative organisms involved. We found that the all-cause mortality on day 14 and day 28 for patients with *P. aeruginosa* were 0% and 14.3%, respectively. On the other hand, the all-cause mortality on both day 14 and day 28 for patients with *A. baumannii* complex were 12.5% (Table 5). All the isolates of DTR-P and CRAB in this study were found to be resistant to meropenem. Although a few isolates of DTR-P were susceptible to ceftolozane/tazobactam, this agent was not used in this patient cohort as the antibiotic was not available due to a supply shortage. Testing of susceptibility to colistin was not performed in any of the cases. Cefiderocol susceptibility was assessed in 18 bacterial isolates. The majority of isolates were found to be susceptible to cefiderocol (13/18, 72.2%); one isolate (5.5%) was determined to have intermediate susceptibility, while four were resistant (22.2%). A total of 11 isolates of *P. aeruginosa* were sent to the reference laboratory managed by the New York State Department of Health for further testing of the presence of carbapenemases; however, none were detected. Furthermore, we noted treatment failure in 2 patients (9.1%).

When analyzed based on age, gender and comorbidities (CCI), no significant differences were found between patients who died after 28 days of cefiderocol therapy and those who survived. Moreover, cefiderocol was generally tolerated well; 2 out of 20 (10%) patients developed mild AKI and 1 out of 12 (8.3%) patients developed mild transaminitis. Notably, two patients were on hemodialysis due to end-stage renal disease at baseline; these patients were excluded from the above calculation.

## 3. Discussion

Cefiderocol has potent in vitro efficacy against carbapenem-resistant Enterobacteriaceae and non-lactose fermenting gram-negative bacilli, such as *P. aeruginosa* (including DTR-P), *A. baumannii* (including CRAB), and *S. maltophilia* [3,6,14]. In the APEKS-cUTI study, cefiderocol was found to be non-inferior to imipenem-cilastatin for treatment of complicated UTI caused by carbapenem-resistant gram-negative organisms [2]. In addition, the CREDIBLE-CR and APEKS-NP study also showed the comparable clinical and microbiological efficacy of cefiderocol to the best available therapy. While the all-cause mortality was found to be lower in patients with cUTI treated with cefiderocol, an increase in all-cause mortality was observed in patients with baseline HABP, VABP and BSI. However, it was not clear whether the higher all-cause mortality is due to the lack of cefiderocol efficacy. In the CREDIBLE-CR study, the all-cause mortality on day 14 and day 28 for all patients was 18.8% and 24.8%, respectively, whereas for patients with BSI, all-cause mortality was 16.7% and 23.3%, respectively. For patients with cUTI, all-cause mortality was 11.5% and 15.4%, respectively, while for patients with HABP/VABP, all-cause mortality was 24.4% and 31.1%, respectively. The APEKS-NP study demonstrated 14- and 28-day all-cause mortality for all patients of 12.8% and 21.2%, respectively [7,11,13]. 

Other case series reported mixed results. Gavaghan et al. reported on outcomes in 24 patients treated with cefiderocol for carbapenem-resistant gram-negative bacterial infections, finding a 30-day mortality of 42%. This case series included patients primarily with pneumonia, as well as CRAB and DTR-P infections, the majority of whom received cefiderocol as monotherapy [15]. Falcone et al. found the use of cefiderocol to be protective in critically ill patients with CRAB infections compared to a colistin-based regimen, with 30-day mortality of 34% and 55.8%, respectively. In this cohort, only 32% received cefiderocol monotherapy [16]. More recently, Bavaro et al. described a more favorable experience in 13 cases of difficult-to-treat gram-negative infections. Microbiological eradication was achieved in all cases, and 30-day mortality was 23%. Every patient received combination therapy with cefiderocol [17].

In contrast, our study found the all-cause mortality to be lower than expected across different categories, such as all-cause mortality for all patients, all-cause mortality based on different types of organisms, or all-cause mortality based on different sites of infections. On day 14 and day 28 for all patients, mortality was 4.5% and 13.6%, respectively. In addition, no significant differences were observed between our study and the 14- and 28-day all-cause mortality results observed in patients treated with cefiderocol monotherapy and double gram-negative coverage, where 14- and 28-day all-cause mortality were 8.3% and 25%, respectively. However, this result was not statistically significant, likely due to the small sample size; thus, more studies should be conducted to better understand the outcomes in those treated with cefiderocol monotherapy compared to combination therapy.

While cefiderocol is recommended by the Infectious Diseases Society of America for treating multidrug-resistant gram-negative infections, combination therapy is advised for moderate-to-severe CRAB and Stenotrophomonas maltophilia infections [18,19]. However, our results build on other reports that suggest there is no significant advantage to using combination therapy with cefiderocol in seriously ill patients [20]. This could have potential important implications in clinical practice from an antibiotic stewardship standpoint, as it could minimize the overuse of unnecessary antibiotics, risks of developing increasing resistance pattern and adverse effects associated with antibiotic exposures. Further study is needed to investigate these aspects to enable potential change in our thinking regarding the use of cefiderocol in clinical practice. Moreover, cefiderocol appeared to be well tolerated by patients in our study and had minimal adverse effects, such as mild AKI and mild transaminitis. This outcome is in alignment with the experiences of other centers [21]. This outcome could also have an important implication for using cefiderocol more widely in clinical practice due to its generally favorable side effect profile.

There are a few limitations in our study. Firstly, the sample size is small and limited to a single center. The data may not be applicable to other patient populations. Secondly, our study was unable to delineate the cause-specific mortality. Mortality rates were only calculated as all-cause mortality. Finally, the retrospective nature of our study was a notable limitation, as it lacks direct comparison with the best available therapy in a randomized double-blind trial fashion.

In conclusion, our findings suggest that cefiderocol could possibly be associated with lower all-cause mortality than previously thought. In our study, we did not find any significant differences between cefiderocol when used in combination with another antibacterial agent and its use as a monotherapy. This finding could have potentially tremendous real-world implications from an antimicrobial stewardship perspective. Further studies are needed to better define the treatment role of cefiderocol in clinical practice.

## 4. Materials and Methods

We performed a retrospective chart review, which included all adult patients (age 18 and above) who received cefiderocol from October 1, 2020 to December 31, 2021 at Stony Brook University Hospital, which is a 624-bed hospital that serves as a tertiary and regional trauma center. Patients who received multiple courses of cefiderocol treatment during the same hospitalization or less than 3 days of cefiderocol treatment were excluded. Those subjects who remained hospitalized as of the end of this study period were also excluded, as the full clinical data were not available for analysis. The primary objective of our study was to assess the 14- and 28-day mortality associated with cefiderocol for infections caused by carbapenem-resistant gram-negative organisms. The secondary objectives were to investigate any associated adverse events, such as worsening renal function, hepatic function and treatment failure requiring change in antimicrobials.

We obtained information on patient variables, including age, Charlson comorbidity index, length of hospitalization, diagnosis, treatment duration, indication, bacterial culture and susceptibility results, site of specimen, minimum inhibitory concentrations (MIC), carbapenemase enzyme production, treatment failure, other antibiotic use and renal and liver function test results. Hospital-acquired infections were identified using CDC surveillance definitions [22]. Organisms were found on various culture specimens, such as blood, urine, sputum/tracheal aspirate, wound and abscess culture. VITEK2 automated system (Biomerieux Inc., Boston, USA) was used for both bacteria identification and susceptibility reporting. Specimens were processed according to the manufacturer’s instructions. Kirby–Bauer disk diffusion (Thermo Fisher Scientific, Waltham, USA) was used for cefiderocol susceptibility testing, as this method was not available on the VITEK2 system. CLSI standards were used to determine antimicrobial breakpoints.

Treatment failure was defined as requiring change in antibiotic from cefiderocol to another antibiotic due to clinical worsening of the patient while taking cefiderocol. Acute kidney injury (AKI) was defined as an increase in serum creatinine by ≥0.3 mg/dL (≥26.5 micromol/L) within 48 h of initiation of cefiderocol or increase in serum creatinine to ≥1.5 times baseline, which was known or presumed to occur within the prior seven days. Worsening hepatic function was defined as an increase in aspartate aminotransferase (AST) and alanine aminotransferase (ALT) levels at the end of cefiderocol treatment by more than 1.5 times compared to levels recorded at the point immediately after which patients started to take the medication. Chi-square and Student’s t-test analyses were performed.

## Figures and Tables

**Table 1 antibiotics-12-00936-t001:** Baseline demographic characteristics of patients.

	Number of Patients (Percentage)
Age range (mean)—years	18–84 (63.6)
Female	7 (31.8)
Male	15 (68.2)
Hypertension	13 (59.1)
Hyperlipidemia	10 (45.5)
Diabetes	10 (45.5)
PVD	5 (22.7)
CVA/TIA/Chronic CNS complications (i.e., subdural hematoma, anoxic brain injury)	6 (27.3)
Chronic kidney disease (CKD)	4 (18.2)
End-stage renal disease (ESRD) on dialysis	2 (9.1)
Chronic heart disease (i.e., CAD, Afib, CHF)	12 (54.5)
Chronic lung disease (i.e., COPD, pulmonary fibrosis, cystic fibrosis, chronic respiratory failure)	7 (31.8)
Chronic liver disease (i.e., cirrhosis, NAFLD, autoimmune hepatitis)	2 (9.1)
Malignancy	3 (13.6)
Organ transplant	2 (9.1)
HIV	1 (4.5)
AIDS	0
Connective tissue disease	1 (4.5)
Charlson comorbidity index score range (mean)	0–11 (5.4)
Length of stay range (mean)—days	11–220 (53)

PVD: peripheral vascular disease, CVA: cerebrovascular accident, TIA: transient ischemic attack, CNS: central nervous system, CAD: coronary artery disease, Afib: atrial fibrillation, CHF: congestive heart failure, COPD: chronic obstructive pulmonary disease, NAFLD: nonalcoholic fatty liver disease, HIV: human immunodeficiency virus, and AIDS: acquired immunodeficiency syndrome.

**Table 2 antibiotics-12-00936-t002:** Types of infection and organisms.

Types of Infection and Organisms	Number of Patients
Lower respiratory tract infection	6
Bloodstream infection	3
cUTI	2
Other sites (intra-abdominal, orbital, and buccal abscess, sacral, lower extremity and facial wounds)	11
DTR-P	14
CRAB	8
*Stenotrophomonas maltophilia*	1

**Table 3 antibiotics-12-00936-t003:** All-cause mortality for all patients, with types of infections and their resistance pattern.

	Number of Patients	14-Day All-Cause Mortality	28-Day All-Cause Mortality	Organisms (Number of Cases)	Pertinent Diagnosis/Clinical Course	Cefiderocol Susceptibility (Number of Cases)	Meropenem Susceptibility (Number of Cases)	Ceftazidime/Avibactam Susceptibility (Number of Cases)	Ceftolozane/Tazobactam Susceptibility (Number of Cases)
Patients receiving cefiderocol	22	1 (4.5%)	3 (13.6%)						
Lower respiratory tract infection	6	1 (16.7%)	1 (16.7%)	CRAB (2) DTR-P (4)	HABP (4) VABP (2)	S (3) I (1) R (1) N/A (1)	R (6)	N/A (4) R (2)	S (1) I (2) R (1) N/A (2)
Bloodstream infection	3	0 (0%)	0 (0%)	CRAB (1) DTR-P (2)		S (2) N/A (1)	R (3)	N/A (2) R (1)	S (1) I (1) N/A (1)
cUTI	2	0 (0%)	0 (0%)	DTR-P (2)		S (1) N/A (1)	R (3)	N/A (2)	S (2)
Other sites (intra-abdominal, orbital, and buccal abscess, sacral, lower extremity and facial wound)	11	0 (0%)	2 (18.2%)						
Intrabdominal abscess				CRAB (2) DTR-P (1)	One patient underwent IR guided drainage 6 days prior to starting cefiderocol	S (3)	R (3)	N/A (2) R (1)	S (1) N/A (2)
Orbital abscess				CRAB (1)	No drainage or surgery performed	S (1)	R (1)	N/A (1)	N/A (1)
Buccal abscess				DTR-P (1)	I&D performed 1 day prior to starting cefiderocol	S (1)	R (1)	R(1)	I (1)
Facial wound				DTR-P (1)	Debridement performed 13 days prior to starting cefiderocol	S (1)	R (1)	R (1)	I (1)
Mid-sternal wound				CRAB (1)	Washout and debridement performed 3 days after starting cefiderocol	S (1)	R (1)	N/A (1)	N/A (1)
Lower extremity wound				CRAB (1) DTR-P (1)	Debridement performed in both patients, 2 days and 3 days before starting cefiderocol, respectively	N/A (1) R (1)	R (2)	N/A (2)	N/A (1) R (1)
Osteomyelitis of left lower extremity digits				DTR-P (1)	Debridement of ulcer and resection of left foot first and second ray performed 2 days after starting cefiderocol	R (1)	R (1)	R (1)	R (1)
Sacral wound				DTR-P (1)	Debridement performed 4 days prior to starting cefiderocol	R (1)	R (1)	R (1)	S (1)

S: susceptible, I: intermediate, R: resistant, N/A: not available, I&D: incision and drainage.

**Table 4 antibiotics-12-00936-t004:** Treatment strategy and all-cause mortality.

Treatment	Number of Patients	14-Day All-Cause Mortality	28-Day All-Cause Mortality	Organisms (Number of Cases)
Combination therapy with cefiderocol	10	0 (0%)	0 (0%)	DTR-P (4) CRAB (6)
Cefiderocol monotherapy	12	1 (8.3%)	3 (25%)	DTR-P (10) CRAB (2)
*p* value		1.0	0.25	

**Table 5 antibiotics-12-00936-t005:** Bacterial isolates and all-cause mortality.

Bacterial Isolates	Number of Patients	14-Day All-Cause Mortality	28-Day All-Cause Mortality
DTR-P	14	0 (0%)	2 (14.3%)
CRAB	8	1 (12.5%)	1 (12.5%)

## Data Availability

Data are contained within the article.

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
