# Peer review of "Clinical Outcome of Cefiderocol for Infections with Carbapenem-Resistant Organisms"

_antibiotics, 2023, doi:10.3390/antibiotics12050936_

Round 1

Reviewer 1 Report

The article is a retrospective study about the use of cefiderocol for patients with Gram-negative infections. Surely, cefiderocol could represent a new option for the treatment of difficult-to-treat infections and its real impact on clinical outcome is still under investigation. 

Authors describe a small cohort of patients with different types of infections caused by different bacteria. Although the work could be helpful in increasing the knowledge regarding the use of cefiderocol, I suggest an extensive review. The reading is difficult to understand, the methods are not well described (such as the definition of the infections, the definition of the inclusion and exclusion criteria, the microbiological tests performed and the resistance pattern of the bacteria involved), as well as the description of the results. 

Here are the detailed comments:

1. The article focuses on the use of Cefiderocol for the treatment of Gram-negative infections. I suggest to better clarify the type of infections and the pathogens involved (ie. cIAI by A. baumannii, VAP by P. aeruginosa etc) and their resistance pattern (i.e. % Carbapenem-Resistant, % Colistin-Resistant). In addition, I suggest clarifying in the “Materials and Methods” section the definition of VAP adopted, since A. baumannii and P. aeruginosa are often colonizing of upper and lower airways, especially in intubated patients.

2. The topic is useful, since Cefiderocol is a new cephalosporin, whose best place in therapy is not yet well defined. Further studies like this are needed to better understand its potential use for difficult infections caused by difficult-to-treat pathogens.

3. There is not a comparison group, and the number of patients included is too small to come o conclusions. For instance, they conclude that the combination therapy is not useful for the treatment of VAP, nevertheless they have a different mortality in the group of combination therapy vs the group of monotherapy. They should discuss this point, because since they analyse only 22 patients, the difference is not significant maybe for the low number of patients.

4.  As mentioned above, methods must be better described. They have to clarify inclusion and exclusion criteria (why excluding patients still hospitalized?) and to describe the definition adopted. In addition, they must describe if patients with cIAI were evaluated by the surgeon and operated on and with what timing. Moreover, they must descrive the microbiological test used (blood cultures, MALDI-TOF, Vitek 2 etc.)

5.  Conclusions are limited by the small number of patients included. Nevertheless, they are in line.

6.  I suggest to cite the work of Bavaro et al., since it’s the first work comparing the use of cefiderocol in different settings with different type of bacteria (Bavaro et al., Antibiotics 2021).

English is difficult to understand, there are lots of grammar mistakes.

Author Response

We thank the reviewer for the comments on our manuscript.  We address the comments as follows:

The article focuses on the use of Cefiderocol for the treatment of Gram-negative infections. I suggest to better clarify the type of infections and the pathogens involved (ie. cIAI by A. baumannii, VAP by P. aeruginosa etc) and their resistance pattern (i.e. % Carbapenem-Resistant, % Colistin-Resistant). In addition, I suggest clarifying in the “Materials and Methods” section the definition of VAP adopted, since A. baumannii and P. aeruginosa are often colonizing of upper and lower airways, especially in intubated patients.

  • The type of infections and pathogens involved, and their resistance pattern are elaborated in table 3. We used CDC surveillance definitions for hospital acquired infections (citation added to the edited manuscript).  

There is not a comparison group, and the number of patients included is too small to come o conclusions. For instance, they conclude that the combination therapy is not useful for the treatment of VAP, nevertheless they have a different mortality in the group of combination therapy vs the group of monotherapy. They should discuss this point, because since they analyse only 22 patients, the difference is not significant maybe for the low number of patients.

  • We have amended the manuscript to note the limitation of small sample size when discussing mortality.  Given the small sample size, we agree that a definitive conclusion cannot be drawn on the benefit of cefiderocol monotherapy versus combination therapy.

As mentioned above, methods must be better described. They have to clarify inclusion and exclusion criteria (why excluding patients still hospitalized?) and to describe the definition adopted. In addition, they must describe if patients with cIAI were evaluated by the surgeon and operated on and with what timing. Moreover, they must descrive the microbiological test used (blood cultures, MALDI-TOF, Vitek 2 etc.)

  • We have amended the manuscript to expand on the Methods section.  We did not include persons who were still hospitalized at the end of the study period as we did not have a full clinical data set to analyze for outcomes.
  • The manuscript was amended with data on those persons with cIAIs to include the surgical interventions that were done and when they were done relative to the initiation of cefiderocol therapy.
  • Microbiological testing for the gram negative isolates was included in the Methods (i.e. VITEK2 for identification and susceptibility testing, Kirby-Bauer disk diffusion for cefiderocol susceptibility testing)

I suggest to cite the work of Bavaro et al., since it’s the first work comparing the use of cefiderocol in different settings with different type of bacteria (Bavaro et al., Antibiotics 2021).

  • We thank the reviewer for alerting us to this reference.  We have incorporated this into the amended discussion.  The Bavaro case series share some similarities to our cohort (overall sick patient populations, comparable mortality outcomes compared to the CREDIBLE-CR and APEKS-NP clinical trials).  While the Bavaro paper describes patients treated only with combination therapy with cefiderocol, we discuss outcomes in persons treated both with cefiderocol as monotherapy as well as in combination.

Reviewer 2 Report

Overall the manuscript is well-written and acknowledges the limitations of the dataset. See comments below which may improve the manuscript.

1. Provide more details regarding failures. What organisms grew in those who died from respiratory tract infections? If the patient was receiving two agents and still experienced mortality, consider stating what the combination was that failed (versus listing what the options were). In the same vein, consider more detail regarding what organisms were treated with combination therapy overall versus monotherapy. This section could have more detail provided.  

2. Did any patient have COVID or how was COVID considered in these patients?  

Author Response

We thank the reviewer for the comments.  We address them as follows:

Provide more details regarding failures. What organisms grew in those who died from respiratory tract infections? If the patient was receiving two agents and still experienced mortality, consider stating what the combination was that failed (versus listing what the options were). In the same vein, consider more detail regarding what organisms were treated with combination therapy overall versus monotherapy. This section could have more detail provided.  

  • We have amended the manuscript to include more details as recommended by the reviewer.  Table 5 includes a breakdown of the organisms associated with the mortality events (all of which occurred in the monotherapy group).
  • CRAB was isolated from the patient’s sputum culture who died from respiratory tract infection/VAP.
  • There was 28-day mortality was not observed in patients who received combination therapy (including cefiderocol) in comparison to cefiderocol monotherapy (0 vs. 25%). However, this was not statistically significant. We suspect that this could be due to the small sample size.

Did any patient have COVID or how was COVID considered in these patients?  

  • None of the patients in this cohort was infected with SARS-CoV-2 during the hospitalization when cefiderocol was prescribed.
  • As per our institution's protocol, SARS-CoV-2 PCR testing was routinely performed on all patients at the time of admission during the study period.

Round 2

Reviewer 1 Report

The manuscript has been modified according to my first report. I suggest a revision of some English mistakes and the modification of the title in “Clinical Outcome of Cefiderocol for Infections caused by Carbapenem-resistant Organisms”. Additionally, I suggest a modification of the layout of table 3 that is difficult to read in its current form.

Some English mistakes are still present.